# Molecular Characterisation of Canine Osteosarcoma in High Risk Breeds

**DOI:** 10.3390/cancers12092405

**Published:** 2020-08-25

**Authors:** Siobhan Simpson, Mark Dunning, Simone de Brot, Aziza Alibhai, Clara Bailey, Corinne L. Woodcock, Madeline Mestas, Shareen Akhtar, Jennie N. Jeyapalan, Jennifer Lothion-Roy, Richard D. Emes, Cinzia Allegrucci, Albert A. Rizvanov, Nigel P. Mongan, Catrin S. Rutland

**Affiliations:** 1School of Veterinary Medicine and Science, Faculty of Medicine and Health Sciences, University of Nottingham, Nottingham LE12 5RD, UK; siobhan.simpson@googlemail.com (S.S.); svzmdd@exmail.nottingham.ac.uk (M.D.); simone.debrot@vetsuisse.unibe.ch (S.d.B.); svaaa3@exmail.nottingham.ac.uk (A.A.); svycb5@exmail.nottingham.ac.uk (C.B.); stxclw@exmail.nottingham.ac.uk (C.L.W.); madeline.mestas@gmail.com (M.M.); saanimalcare@gmail.com (S.A.); plzjnj@exmail.nottingham.ac.uk (J.N.J.); stxjl34@exmail.nottingham.ac.uk (J.L.-R.); svzrde@exmail.nottingham.ac.uk (R.D.E.); mgzca@exmail.nottingham.ac.uk (C.A.); rizvanov@gmail.com (A.A.R.); catrin.rutland@nottingham.ac.uk (C.S.R.); 2Willows Veterinary Centre and Referral Service, Solihull B90 4NH, UK; 3COMPATH, Institute of Animal Pathology, University of Bern, Laenggassstrasse 122, CH-3012 Bern, Switzerland; 4Biodiscovery Institute, Faculty of Medicine and Health Science, University of Nottingham, University Park, Nottingham NG7 2RD, UK; 5Institute of Fundamental Medicine and Biology, Kazan Federal University, Kazan 42008, Russia; 6Department of Pharmacology, Weill Cornell Medicine, New York, NY 10065, USA

**Keywords:** androgen, androgen receptor, bone, cancer risk, canine, KEGG, Wiki

## Abstract

Dogs develop osteosarcoma (OSA) and the disease process closely resembles that of human OSA. OSA has a poor prognosis in both species and disease-free intervals and cure rates have not improved in recent years. Gene expression in canine OSAs was compared with non-tumor tissue utilising RNA sequencing, validated by qRT-PCR and immunohistochemistry (*n* = 16). Polymorphic polyglutamine (polyQ) tracts in the androgen receptor (*AR/NR3C4*) and nuclear receptor coactivator 3 (*NCOA3*) genes were investigated in control and OSA patients using polymerase chain reaction (PCR), Sanger sequencing and fragment analysis (*n* = 1019 Rottweilers, 379 Irish Wolfhounds). Our analysis identified 1281 significantly differentially expressed genes (>2 fold change, *p* < 0.05), specifically 839 lower and 442 elevated gene expression in osteosarcoma (*n* = 3) samples relative to non-malignant (*n* = 4) bone. Enriched pathways and gene ontologies were identified, which provide insight into the molecular pathways implicated in canine OSA. Expression of a subset of these genes (*SLC2A1*, *DKK3*, *MMP3*, *POSTN*, *RBP4*, *ASPN*) was validated by qRTPCR and immunohistochemistry (MMP3, DKK3, SLC2A1) respectively. While little variation was found in the *NCOA3* polyQ tract, greater variation was present in both polyQ tracts in the *AR*, but no significant associations in length were made with OSA. The data provides novel insights into the molecular mechanisms of OSA in high risk breeds. This knowledge may inform development of new prevention strategies and treatments for OSA in dogs and supports utilising spontaneous OSA in dogs to improve understanding of the disease in people.

## 1. Introduction

Osteosarcoma (OSA) is a rare cancer that typically affects adolescents and the elderly [1], and survival rates for OSA have seen little improvement in more than 20 years [2,3]. One challenge to the development of improved treatments for OSA in people is the limitation of relevant models. Animal models of OSA rely on chemical induction, xeno/allografts, and genetically engineered animals that do not fully reflect aspects of spontaneously occurring disease [4,5,6,7]. Canine OSA occurs spontaneously and is believed to share similarities with human OSA [4]. Published incidence rates are 27.2 dogs and 0.89 people per 100,000 population [8,9]. Aside from potentially improving the understanding of OSA in people, canine OSA is a significant veterinary clinical challenge in its own right. One-year survival rates following diagnosis and treatment are typically lower than 45% [10,11,12].

Risk factors for people developing OSA include sex, race, puberty, and growth [3,13,14]. Canine OSA typically affects large and giant breeds, with Irish Wolfhounds (IWH), Rottweilers, Scottish Deerhounds, St Bernards, and Great Danes particularly affected [9], reflecting the human population where individuals with OSA are more likely to be taller than average [9,14,15]. In the canine population, as with human populations, there also appears to be a skewed sex ratio with males more likely to develop OSA than females [1,9]. Neutering status, although less relevant in the human context, also appears to contribute to OSA risk, where neutered dogs are more likely to develop OSA than those that remain entire [16]. This, combined with the association with puberty, suggests a complex role for sex hormone signalling in OSA risk.

The androgen receptor (AR/NR3C4) is a member of the ligand-dependent superfamily of transcription factors [17,18,19,20]. In the presence of an agonist, the AR recruits many epigenetic coregulators to mediate transcriptional regulation [21]. Notable examples of coregulators include the p160 coactivators, NCOA1, NCOA2 and NCOA3 [21]. The *AR* gene is X-linked, with both males and females possessing functional *AR*, however, differing levels of circulating androgens between the sexes result in different receptor activation [22]. Mutations in the *AR* gene can alter the sensitivity of the receptor to androgens [23,24]. Some *AR* mutations cause complete androgen insensitivity where genetically 46XY males appear to be phenotypically female [18,23,25,26]. In addition to loss of function *AR* mutations, there is a polymorphic polyglutamine (polyQ) tract within the *AR* gene that shows variation in the polyQ repeat number [27,28]. The variation in *AR* gene polyQ repeats has been associated with prostate cancer, breast cancer, ovarian hyperandrogenism, and Kennedy’s disease [29,30,31,32,33]. Individuals with shorter polyQ tracts are more sensitive to androgens, while individuals with longer polyQ tracts are less sensitive [27,28]. Androgen activation of the *AR* requires the presence of coactivators [34,35,36,37]. One such coactivator, *NCOA3*, also harbours a polyQ repeat [38,39]. Expression of NCOA3 has been associated with increased AR activation in prostate cancer and urothelial carcinoma of the bladder in humans [35,37]. In addition, increased expression of NCOA3, with implications for *AR* activation, and variation in the length of the polyQ repeat in the *NCOA3* gene have been associated with bone cancer, epithelial ovarian cancer, colorectal cancer, and breast cancer in humans [40,41,42,43]. The polyQ tract within the *AR* and *NCOA3* genes has not been investigated in relation to canine OSA.

Although OSA itself does not generally appear to be heritable, the risk factors themselves have heritable components [44]. To date there have been no external modifiable risk factors identified in the development of OSA outside of individuals with inherited cancer syndromes, where radiation has been identified as a factor in tumor development [45,46]. In people, some rare heritable syndromes have been found to increase the risk of developing OSA [47,48,49]. The lack of identified modifiable risk factors limits the development of effective OSA prevention strategies, thus to date the emphasis has been placed on the development of early diagnostic and improved treatment approaches to advance OSA outcomes in people. Over 900 genes are associated with human OSA [50], but only two somatic genetic mutations have been specifically associated with OSA [13]. This lack of identified genetic associations is not surprising considering the lack of heritability observed in human OSA. Genomic and chromosomal instability is a reported factor in many types of cancer progression [51,52], and OSA has been shown to display chromosomal instability associated with mutations in the *TP53* gene [53]. A consequence of this chromosomal instability is aneuploidy, which can lead to the overexpression of some genes within malignant cells, disrupting normal cell processes [54]. Although mutations in *TP53* appear to be associated with chromosomal instability, the gene itself does not seem to be subsequently over expressed following aneuploidy [53,54].

In contrast to human OSA, canine OSA may be heritable, with an apparent predisposition in some breeds [9,55]. Interestingly, of the 15 breeds with the highest reported incidence of OSA, 12 are within a unique clade on the canine phylogenetic tree [9,56]. This relationship between affected breeds could indicate a potential common genetic origin of canine OSA; however, the clade is large and also contains many breeds that do not commonly develop OSA [56]. There have been 34 genetic loci associated with canine OSA across four breeds [57,58]. None of the loci are consistently associated across breeds, further suggesting that there may be a difference between breeds in the genetic predisposition to developing canine OSA [57,58]. Currently none of the genetic variants identified as associated with canine OSA have had their mechanism of action verified. Differentially expressed genes between canine OSA and non-tumor tissue have been identified that have implications for growth and metastasis, are potential drug targets [59,60,61,62], and are associated with survival time [63,64,65,66]. Additional work is required to confirm the effect of the genetic loci identified as associated with canine OSA, and to account for the variation observed in the development of disease between breeds [57,58]. The Irish Wolfhound (IWH) breed has the highest prevalence of OSA [9] with one of the lower median ages of onset at 6.6 years [9], with four loci associated with OSA development [58]. The IWH is closely related to the Scottish Deerhound as the latter breed was used to re-establish the breed in the early 1860s through to the 1930s [67]. Scottish Deerhounds are a minority breed and therefore do not commonly appear on incidence breed lists, but OSA is a common problem within this breed with an incidence rate of more than 15%, double that of the general canine population [16,55]. Heritability has been estimated at 0.69; both dominant and recessive models have been suggested [55,68]. One study has shown linkage between OSA and chromosome 34q16.2–17.1; a region syntenic to human chromosome 3q26, which is also associated with OSA incidence [57]. Rottweilers have the fifth highest incidence ranking for OSA with a median age of onset of 7.9 years [9] and 15 OSA associated loci [58].

Improving OSA survival rates will only be possible with additional understanding of the disease and the development of novel treatments and drugs targeting OSA-specific pathways [69,70,71]. The present study used next generation RNA sequencing to identify novel differentially expressed genes in OSA tumor tissue as compared to matched non-tumor tissue and assessed *NCOA3* and *AR* polyQ tract variations in affected and non-affected IWH and Rottweiler dogs.

## 2. Results

### 2.1. RNAseq Gene Expression

The Illumina iGenomes CanFam3.1 Ensembl (Illumina, San Diego, CA, USA) annotated genome comprising 24,580 genes was included in the RNAseq analysis. Of these genes, 1281 were identified with a fold change of ≥2 between tumor and non-tumor tissue and a *p* < 0.05 (*n* = 4 non-tumor and *n* = 3 tumor samples). Higher expression of 442 genes and lower expression of 839 genes was observed in tumor tissue compared to the non-tumor tissue (Figure 1A,B, Table 1 and Table 2, and Appendix A). The biological relevance of these significantly differentially expressed genes was analysed using over-representation of enriched pathways, gene ontologies and predicted transcription factor networks using the WebGestalt platform (http://www.webgestalt.org/, Zhang Lab, Baylor College of Medicine, Houston, TX, USA; Appendix A). Differentially expressed genes were significantly associated with pathways associated with heme biosynthesis, calcium homeostasis and signalling (Table 2 and Table 3). Significantly enriched gene ontologies and enriched predicted transcription factor networks were also identified (Appendix A).

### 2.2. qRT-PCR and Immunohistochemistry Validation

Validation of seven differentially expressed genes identified by RNAseq was performed by qRT-PCR analysis of eight non-malignant bone specimens and seven OSA specimens. Expression for *MMP3* (*t* = 4.884, *p* < 0.0001), *SLC2A1* (*t* = 3.703, *p* = 0.0006), *DDK3* (*t* = 3.981, *p* = 0.0003), *POSTN* (*t* = 2.061, *p* = 0.0455), *RBP4* (*t* = 3.048, *p* = 0.004) and *ASPN* (*t* = 2.733, *p* = 0.0091) was confirmed to be higher in the tumor tissues (Figure 2A–G). Differential expression of *S100A8* was not confirmed by qRT-PCR (*t* = 1.766 *p* = 0.0847; Figure 2H). Immunostaining confirmed positive expression of MMP3, SLC2A1 and DKK3 proteins in OSA tissue (Figure 3).

### 2.3. Cohort Epidemiology

From the buccal swab cohort, 12 IWHs (8 females and 4 males) and 31 Rottweilers (19 females and 12 males) were reported with OSA (diagnosed by veterinary surgeons and reported by owners), representing 0.53% of all IWHs (from *n* = 379) and 3.04% of Rottweilers (from *n* = 1019) within the entire Nottingham cohort inclusive of dogs of all ages. Significant differences were identified in the age at OSA diagnosis between males and females in both breeds. Mean age at OSA diagnosis for male IWHs was 5.34 years, whereas in females this was 7.65 years (*t* = 3.04, *p* = 0.0069). A similar difference in the age at diagnosis was observed in Rottweilers, with a mean age at diagnosis for males of 7.00 years while that for females was 8.50 years (*t* = 2.34, *p* = 0.027, summarised in Table 4). There were no differences between the sexes in the proportion of individuals affected by OSA in IWHs—χ^2^test result = 0.065 (*p* = 0.80), or Rottweilers—χ^2^test result = 0.17 (*p* = 0.68).

Of IWH males, 80% that developed OSA had done so before the age of 6.5 years, while 80% of IWH females that developed OSA had done so by 9 years. Rottweilers tend to live longer than IWHs, therefore a 90% threshold was used to determine the age restrictions in this breed. Of male Rottweilers, 90% of those that developed OSA were diagnosed before the age of 9 years, whereas 90% of OSA-affected females were diagnosed before the age of 10 years. Using these age restrictions, there were 23 male IWHs unaffected by OSA over 6.5 years and 6 female IWHs unaffected by OSA over 9 years from the cohort of 379. There were 51 male Rottweilers unaffected by OSA over the age of 9 years and 53 female Rottweilers unaffected by OSA over 10 years old from the cohort of 1019. These individuals were used as control samples for polyQ tract analysis (summarized in Table 5).

### 2.4. Androgen polyQ Analysis

Across all individuals genotyped at both *AR1* and *AR2*, the total length of the *AR* polyQ tract ranged from 33 to 35 repeats. The length of the *AR1* polyQ tract ranged from 10 to 12 repeats and the length of the *AR2* polyQ tract ranged from 23 to 25 repeats (Figure 4). *NCOA3* had a range of 15 to 16 repeats within the polyQ tract across all genotyped individuals (Figure 4). There were no significant differences in mean *AR* polyQ repeat length between IWHs (*t* = 2.04, *p* = 0.053) or Rottweilers (*t* = 1.00, *p* = 0.33) with and without a diagnosis of OSA. When individuals were split by sex there remained no significant differences between groups (IWH OSA: ANOVA—F = 1.28, *p* = 0.30; Rottweiler OSA: ANOVA—F = 1.88, *p* = 0.14, Table 5). Similar results were obtained when *AR1* and *AR2* were considered separately. There were also no significant differences in *AR1* or *AR2* length between individuals with and without OSA in IWHs (*AR1 t* = 1.02, *p* = 0.33; *AR2 t* = 1.83, *p* = 0.08) and Rottweilers (*AR1 t* = 1.00, *p* = 0.33; *AR2 t* = 1.00, *p* = 0.32). The length of *AR1* or *AR2* did not differ significantly when sex was compared in IWHs (*AR1* F = 0.80, *p* = 0.50; *AR2* F = 1.04, *p* = 0.39; Table 5) or Rottweilers (*AR1* F = 1.88, *p* = 0.14; *AR2* F = 0.55, *p* = 0.65; Table 5).

There were no significant differences in mean *NCOA3* polyQ repeat length between IWHs with a diagnosis of OSA and IWHs included in the unaffected by OSA group, as all individuals had 15 repeats in their polyQ tract. There were two IWH individuals in the study that had 16 repeats; these individuals were unaffected by OSA but had not yet reached the age restrictions to be included in the unaffected group analysis. Similarly, there was little variation in the *NCOA3* polyQ repeat length in Rottweilers with two individuals in the unaffected by OSA group having a mean repeat length of 15.5. The resulting *t*-test was not significant (*t* = 1.42, *p* = 0.16).

Analysis of the frequencies of the entire populations demonstrated significant differences in *AR1* and *AR2* polyQ repeat frequencies between the two breeds (*AR1* χ^2^test result = 1707, *p* = < 0.00001; *AR2* χ^2^test result = 1687; *p* = <0.00001). IWHs had fewer repeats at *AR1*, but more repeats at *AR2*, compared to Rottweilers (4). The mean repeat length overall of *AR* was significantly increased in IWHs at 34.25 (range 33–35.5) repeats in comparison to Rottweilers at 34.02 (range 33–35; *t*-test, *t* = 10.62, *p* = <0.0001). There was no significant difference between breeds in the *NCOA3* polyQ repeat frequencies, χ^2^test result = 1.79, (*p* = 0.18).

## 3. Discussion

Differential gene expression in cancers has been associated with disease progression and prognosis [70,72,73,74]. This study identified 583 genes as differentially expressed between canine OSA (*n* = 3) and non-tumor (*n* = 5) bone tissue by RNAseq analysis (Figure 1). Many of these genes had previously been identified as differentially expressed in other cancers and could be potential drug targets in the treatment of OSA. qRT-PCR was used to validate RNAseq results for *MMP3 SLC2A1*, *DKK3*, *POSTN*, *RBP4,* and *ASPN* (Figure 2). Separately we used immunohistochemistry to confirm protein expression and localisation of MMP3, SLC2A1 and DKK3 in bone and OSA specimens (Figure 3).

The *MMP3* gene encodes a member of the matrix metalloproteinase proteins, which degrades a range of extracellular matrix components and is therefore an important mediator of metastatic invasion [75,76]. High MMP3 protein expression has been shown in breast, lung, and pancreatic cancer, and has been associated with poor prognosis [77]. In addition, *MMP3* expression in the murine model of mammary carcinoma has been demonstrated to be important for primary tumor and metastasis growth [78]. Formation of metastases in OSA is a critical stage in disease progression associated with poor prognosis [79,80]. A selective inhibitor of MMP3 is available (UK370106) but is yet to be tested for inhibition of primary tumor or metastatic tumor growth [81]. A generic MMP inhibitor is available (Marimastat, Sigma-Aldrich, St. Louis, MO, USA) [82], but it was not effective in clinical trials [82,83,84,85]. In pancreatic small-cell lung cancers trials, it was not established whether the tumours expressed high levels of *MMP*s, which could have had implications in establishing drug efficiency [82,85]. Marimastat has not been tested in OSA. In the present study, it was shown that expression of *MMP3* is elevated in canine osteosarcoma in comparison to non-malignant bone tissue. Selective metalloprotease inhibitors such as UK370106 and/or marimastat and related candidate drugs may improve the prognosis of canine OSA in those cases with elevated MMP3 expression. Previously published studies found that *MMP3* was more highly expressed in OSA tumors than in normal bone [86,87], which corroborated the results found in this study.

Glucose is an essential component in cellular metabolism, with transport into cells reliant on membrane glycoproteins [88]. *SLC2A1* is a gene encoding one such cell membrane glycoprotein, which has been shown to be involved in glucose transport in a range of tumors including oral squamous cell carcinoma, non-small cell lung carcinoma, and non-gastrointestinal stromal tumor soft tissue sarcomas [89,90,91]. In patients with OSA, higher expression of *SLC2A1* within tumors has been associated with a shorter disease-free interval, and poorer prognosis than in patients with lower expression of *SLC2A1* [92]. In the current study, *SLC2A1* expression was significantly higher in OSA tissue compared to non-tumor tissue. Survival time was not available for patients involved in the current study, but canine OSA survival time is typically short, up to 14.4 months on average [93]. Based on the current study results combined with those previously published [92], this could be due to the high expression of *SLC2A1* [10,11,12]. There are several drugs available that have inhibitory effects on cellular glucose transport and *SLC2A1*, but few have been tested as a treatment for OSA [94,95,96,97].

A range of cancers have been shown to have aberrant Wnt activity [98,99,100,101]. Dickkopf proteins inhibit Wnt signalling and have been shown to be differentially expressed in colon cancer, prostate cancer, and breast cancer [102,103,104]. Reduced *DKK3* expression has been associated with endometrial cancer, cervical cancer and breast cancer, and as such it has been implicated as a tumor suppressor [99,105,106,107]. Within OSA there are conflicting reports; in OSA cell lines and xenograft mice, *DKK3* expression was shown to be downregulated, with subsequent restoration of *DKK3* expression reducing tumor and metastatic growth [108]. In contrast, *DKK3* has been shown to be more highly expressed in OSA cells that overexpress *NKD2* [109]. The results from the current study showed that *DKK3* was more highly expressed in OSA tissue than in non-tumor tissue. This is in contrast to expression in most other cancer types, but in agreement with other findings [109]. Knockdown of *DKK3* in cells overexpressing *NKD2* increased their proliferative potential indicating that *DKK3* expression could be mediating *NKD2*-induced metastasis [109]. There are currently no drugs available that act on DKK3. MMP3 and DKK3 have been shown to have opposite effects on Wnt activation. MMP3 secretion has been shown to lead to increased Wnt activation [110,111], while increased *DKK3* expression results in reduced Wnt activation [99,105,106,107]. The results presented in the current study thus appear to contradict these earlier findings, with both *MMP3* and *DKK3* expressed more highly in tumor compared to non-tumor tissue. It is possible that DKK3 is ameliorating the effect of MMP3 secretion in the tumors in this study in a similar way to the effect identified by Zhao and colleagues [109]. Further work is required to establish the relationship between *MMP3* and *DKK3* in OSA.

To assess the functional relevance of the differentially expressed genes identified here, we performed a comparative search of the KEGG, Panther and Reactome pathway databases using the WebGestalt platform (www.webgestalt.org; Table 3). By integrating the results obtained from these complementary databases, the most detailed insight into the mechanistic relevance of the differentially expressed genes can be obtained. Interestingly, this approach identified aberrant expression of genes associated with heme biosynthesis. While the exact role of heme in cancer is controversial [112], aberrant heme metabolism may influence tumor energy metabolism, the tumor microenvironment, angiogenesis and metastases [112]. Expression of the nine genes (*ALAD*, *ALAS1*, *COX10*, *CPOX*, *FECH*, *HMBS*, *NFE2L1*, *UROD*, *UROS*) of the heme biosynthesis ontology were lower in OSA as compared with non-malignant bone. This could be attributed to the loss of normal bone function as bone marrow is a major source of heme synthesis and hematopoiesis [113]. Bone marrow space may have been replaced with tumor tissue with a resulting loss in functional marrow. An additional possibility is that the loss of heme synthesis within the tumor contributes to its progression. There is some evidence that tetrapyrroles, such as heme, possess anti-cancer properties by inducing apoptosis [114]. Biliverdin and bilirubin, the catalytic products of heme generated by heme oxygenases, have been shown to have anti-oxidant properties [115]. Due to the reduced expression of genes involved in tetrapyrrole/heme/pigment metabolic and biosynthetic processes in tumor compared to non-tumor tissue in this study, it would seem likely that the apoptosis and anti-oxidant properties of tetrapyrroles is correspondingly reduced.

Pathways associated with calcium signalling and cardiomyopathy (Table 1) involved intracellular signalling enzymes including integrins, adenylate cyclases 4 and 9, ATPases, and calcium channels, and indeed calcium homeostasis may reflect aberrant osteoblastic differentiation during osteosarcoma carcinogenesis [116]. Consistent with this, multiple gene ontologies related to cellular differentiation, morphogenesis, development, cellular proliferation, and metabolism were associated with the differentially expressed genes identified here (Appendix A).

We used the WebGestalt network module (www.webgestalt.org) to search the human MSigDB transcription factor target database. This analysis identified 51 transcription factor regulators (Appendix A) predicted to act as regulators of OSA-associated differentially expressed genes (Appendix A). Notable predicted regulators included LEF1, a key regulator of the Wnt/β-catenin pathway, which has previously been implicated in OSA [117,118]. There were 122 putative LEF1-target genes identified in the differentially expressed genes. *LEF1* expression was significantly different between tumor and non-tumor tissue (*p* = 0.0405) with higher expression seen in tumor compared with non-tumor tissue. Among the genes regulated by *LEF1,* some demonstrated increased expression in tumor compared with non-tumor tissue, whereas others showed decreased expression. Genes do not act in isolation, therefore, those genes with lower expression in this dataset may have also required the expression of additional transcription factors or coactivators for activation [119]. *LEF1* is associated with increased Wnt signalling, however, in the current study it appears that genes may be antagonising one another, with genes that both increase and decrease Wnt signalling expressed more highly in tumor compared with non-tumor tissue [109,110,111,120]. Wnt signalling has both canonical and non-canonical pathways, with non-canonical signalling split into planar cell polarity and Wnt/Ca^2+^ [121].

Expression of DKK3 and MMP3 are increased in OSA (Figure 2) and both are known to act on the binding of Wnt activator proteins so alterations in the amount of these proteins will affect both canonical and non-canonical Wnt signalling [110,121,122]. Combined with the increase in *DKK3* and *MMP3* expression seen in tumor tissue, the demonstration of aberrant activation of *LEF1* in the current study adds OSA to the range of cancers that exhibit aberrant Wnt activity [98,99,100,101]. *LEF1* is part of the canonical Wnt signalling pathway, requiring β-catenin as a coactivator for downstream gene transcription [123,124]. It may be that aberrant Wnt processes upstream of *LEF1*, such as *DKK3* and MMP3 expression, are affecting *LEF1* activity. This may explain the difference in the transcription of genes regulated by *LEF1*, but not *LEF1* itself in OSA tissue compared to non-tumor tissue. *LEF1* could be an additional drug target to influence Wnt signalling within OSA. In addition to association with *CTGF,* it has been shown that expression of *MMP3* is inversely related to expression of *IRF8* [78]. Due to this, IRF8 could also be an alternative drug target.

In total, 0.53% of all IWHs (*n* = 379) and 3.04% of Rottweilers (*n* = 1019) had OSA and although there were no overall differences in the rates of males and females affected, the males were younger at diagnosis than females, with a reduction of 2 years and 1.5 years seen for IWH and Rottweiler males, respectively. It is possible that other factors play a role in this. For example, IWH males are more likely to die younger from dilated cardiomyopathy and atrial fibrillation than females [125,126]. As heart disease is the most common specific cause of death in this breed, males may not always live long enough to develop OSA symptoms/get a diagnosis. The 379 IWHs used in this study were also used in previous publications to look at population outcomes in cardiomyopathy and atrial fibrillation [126]. The results showed that 65% of the male dogs with dilated cardiomyopathy and/or atrial fibrillation had been diagnosed before the age of 5.5 years—the average age at which OSA was diagnosed, whereas only 40% of females had been diagnosed as suffering from these heart diseases at that age [126].

This is the first study to examine the length of the canine *NCOA3* polyQ tract. This is also the first evaluation of the *AR* polyQ tract in IWHs and Rottweilers. Most individuals had 15 polyQ repeats in both alleles of the *NCOA3* gene, with only one additional repeat in 1% of Rottweiler and 0.5% of IWH *NCOA3* polyQ tracts genotyped. The lack of variation in the length of the *NCOA3* polyQ tract in these two breeds strongly suggests that variation in the length of the *NCOA3* polyQ tract is not a contributing factor in the development of OSA in IWHs, or Rottweilers. The length of the canine *AR* polyQ tract has previously been associated with aggression in male Japanese Akita Inus and prostate cancer in a variety of breeds [127,128]. In the current study, there was no association between the length of the canine *AR* and OSA in either IWHs or Rottweilers. Additionally, RNAseq did not identify differential expression of the *AR* or *NCOA3* in canine OSA relative to non-tumor bone tissue. It was interesting that IWHs and Rottweilers had significantly different allele frequencies at both *AR1* and *AR2,* which supports previous findings across other breeds [129]. This difference in allele frequencies may be affected by different population histories, with founder effects, inbreeding, and genetic hitch-hiking [130,131,132].

## 4. Materials and Methods

### 4.1. Samples and Ethics

This study was approved by the University of Nottingham ethics committee (numbers 1823 160714 and 959 130925, 20 July 2016) in compliance with Home Office regulations and the Veterinary Surgeons Act. The call for owners to participate in research was made to the public and veterinary practices via breed health group collaborations, emails, social media, attending breed shows, and by holding breed health information days in-house. Informed consent was obtained from all dog owners.

Canine OSA tumor (*n* = 7) and non-tumor (*n* = 8) tissue was obtained from veterinary surgeons following amputation treatment or immediately following euthanasia. Confirmation of OSA was confirmed by pathologists via histopathological examination. A 1 cm^3^ piece of tumor was extracted along with a 1 cm^3^ piece of adjacent non-tumor affected bone tissue. These samples were placed directly into RNA*later*^®^ (Sigma-Aldrich, St. Louis, MO, USA), shipped at room temperature and stored at −20 °C.

DNA samples from 1019 Rottweilers and 379 IWHs were collected using Isohelix DNA Buccal Swabs (Cell Projects Ltd., Harrietsham, UK), according to the manufacturer’s instructions. Clinical histories of each animal were also obtained from owners. Swabs and clinical information were collected over a 3-year period with regular updates checked for a further 2 years. A Chi-Square (χ^2^test) was performed to establish any significant difference in the number of males and females diagnosed with OSA. In addition, *t*-tests were performed using R statistical software [133] to compare the mean age of diagnosis between males and females. From this, appropriate age restrictions for each sex were established for the inclusion of unaffected individuals in the control group, based on the age by which most affected individuals had been diagnosed. Individuals in the unaffected group were used in genetic association testing.

### 4.2. RNA Extraction and Next Generation Sequencing

OSA tumor (*n* = 3) and non-tumor (*n* = 4) bone tissue samples were homogenized (GentleMACS, Miltenyi Biotec, Bergisch Gladbach, Germany) and RNA extracted using Qiagen RNeasy Mini Kit (Qiagen, Hilden, Germany) according to manufacturer’s instructions. The extracted RNA was stored at −80 °C. RNA was quantified using both a NanoDrop 8000 (Thermo Fisher Scientific, Waltham, MA, USA) and an Agilent 2100 Bioanalyzer (Agilent, Santa Clara, CA, USA). RNAseq analysis of seven samples comprised of four non-malignant bone and three OSA specimens and was completed at Edinburgh Genomics using an Illumina HiSeq platform (Illumina, San Diego, CA, USA) with ~20 million reads obtained from all samples analysed. The resultant raw fastq reads were processed for quality (phred score >30) and adapter sequences removed using Trim Galore (https://www.bioinformatics.babraham.ac.uk/projects/trim_galore/). The processed raw reads were aligned to the canine genome (CanFam3.1, https://www.ncbi.nlm.nih.gov/assembly/GCF_000002285.3/) using the Ensembl annotated iGenomes build using Tophat2 [134]. Differential gene expression analysis was completed using Cufflinks/Cuffdiff [135] with significantly differentially expressed genes between the tumor and non-tumor tissue showing a fold change ≥2, and *p* < 0.05. WebGestalt [136] was also used to perform comparative gene ontology, KEGG, Panther and Reactome pathway analysis, and enriched transcription factor target networks to establish insights into the molecular mechanisms of canine OSA [137,138]. To do this, gene symbols were used to search the human pathway and network databases accessed via WebGestalt. A subset of these genes was selected for qPCR validation based on implication in cancer and biological function. RNA sequences in fastq format and gene quantification values as determined by Cufflinks/Cuffdiff expressed in FRPM have been deposited in the NCBI GEO database (accession = GSE155646, https://www.ncbi.nlm.nih.gov/geo/).

### 4.3. Quantitative Reverse Transcriptase PCR

cDNA was generated from the RNA extracted from OSA tumors and associated non-tumor tissue by reverse transcriptase PCR. The reverse transcriptase PCR reaction mixture consisted of 2 µL of 5× qScript™ cDNA SuperMix (Quanta Biosciences, Beverly, MA, USA), 0.5–1 µg RNA, adjusted to 10 µL using PCR grade H_2_O. Reverse transcriptase PCR was carried out at 25 °C for 5 min followed by 1 h at 42 °C. Gene expression of *MMP3*, *MMP2*, *SLC2A1*, *DKK3*, *POSTN*, *RBP4*, *ASPN,* and *S100A8* were quantified using qRT-PCR using TaqMan assays Cf02625966_g1, Cf02649247_g1, Cf02633301_m1, Cf02633587_m1, Cf02634869_m1, Cf02659289_m1, Cf02655953_m1 respectively, ThermoFisher Scientific, Waltham, MA, USA) as described previously [139]. Expression for all samples was normalised to the internal control Actin (Taqman assays Cf03023880_g1, ThermoFisher Scientific, Waltham, MA, USA) and relative expression of tumor compared to non-tumor tissue was calculated utilising the methods described by Pfaffl [140]. *t*-tests were performed using GraphPad Prism version 7.01 for Windows (GraphPad Software, La Jolla, CA, USA).

### 4.4. Immunohistochemistry

Immunohistochemistry was performed to show positive protein expression of a subset of the genes analysed, SLC2A1, MMP3 and DKK3. Rottweiler post-mortem OSA tissue (*n* = 16) was obtained from Bridge Pathology, UK in the form of OSA tissue with neighbouring non-OSA bone. Tissue was fixed in 4% paraformaldehyde for 2 h, dehydrated through an ethanol series, embedded into paraffin blocks, and sectioned at 7 μm. Immunohistochemistry was carried out using a Leica Novolink Polymer Detection Kit (Leica, Wetzlar, Germany) according to manufacturer’s protocols with primary antibodies diluted in fetal calf serum 1:100; anti-SLC2A1 polyclonal unconjugated rabbit antibody (100732-TOB-SIB; Stratech, Ely, UK), anti-MMP3 polyclonal unconjugated rabbit antibody (GTX74514; GeneTex, Irvine, CA, USA), anti-DKK3 (N1C3) polyclonal unconjugated rabbit antibody (GTX100571; GeneTex, Irvine, CA, USA) were used to stain proteins of interest. Microscopy was carried out to confirm positive staining cytoplasmic and/or nuclear staining (Leica, Wetzlar, Germany). Negative controls received no primary antibody and were incubated in fetal calf serum only.

### 4.5. PolyQ Analysis

DNA extraction was performed as previously described [126]. PCR and fragment length analysis: The canine *AR* harbours two polyQ repeat regions within its locus, hereby denoted *AR1* and *AR2* [129]. *AR1* and *AR2* were PCR amplified using primers published by Maejima et al. [129]. Primers were designed to flank the polyQ region of the canine *NCOA3* gene. One of each pair of primers was fluorescently labelled with HEX or 6-FAM dyes (Sigma-Aldrich, St. Louis, MO, USA) to allow fragment analysis to be carried out (Primers: *AR1* forward 6-FAM-CCGTGAGCGCAGCACCTCCCGGTG, reverse AGGCTGACCGCTGTTGGGAAGGCTGC; *AR2* forward 6-FAM-GCCAGCACCACCGGACGAGAATGA, reverse TAACTGTCCTTGGAGGAGGTGGAAGCA; *NCOA3* forward HEX-CCCAGCAGGGTTTTCTGAATGCCC, reverse CACAGGCCCTGCCAAAACGCCATCC). Pre-PCR multiplexing, whereby two PCRs are carried out on a sample as a single reaction, was performed for *AR1* and *NCOA3* with no discernible effect on PCR efficiency. The reaction mixtures consisted of 1× LightCycler^®^ 480 Probes Master Mix (Roche, Basel, Switzerland), 0.5 µM of primers, and 1.5 µL of template DNA, with the volume adjusted to 15 µL using PCR grade H_2_O. The calculation for a 15 µL reaction containing *AR1* and *NCOA3* primers combined was 7.5 µL 2× Roche Probe Master Mix, 1.5 µL of each primer from a stock with concentration of 5 µM, and 1.5 µL template DNA. *AR2* primers were added to the multiplex post PCR; the calculation for a 15 µL reaction for the *AR2* primer pair was 7.5 µL 2× Roche Probe Master Mix and 1.5 µL of each primer from a stock with concentration of 5 µM, 3 µL H_2_O, and 1.5 µL template DNA. The PCR reaction was run for 40 cycles at 94 °C for 30 s, annealing temperature for 30 s, and 72 °C for 30 s. The optimal annealing temperature was determined through a temperature gradient PCR, with temperature ranging between 52 °C and 64 °C. An annealing temperature of 61 °C was used for *AR1* and *NCOA3*, and 57 °C was used for *AR2*.

All multiplex samples were diluted by a factor of 100. Analysis of the fluorescently-labelled PCR products was undertaken using the Applied Biosystems 3730 DNA Analyzer with GeneScan ROX-500 size standard (DBS Genomics, Durham, UK) for each sample. The size discrepancy between *AR1* and *AR2* allowed for discrimination between their FAM-labelled products. Genotypes were scored using Genemapper software version 3.7 (Applied Biosystems, Foster City, CA, USA). A homozygote of each allele was Sanger sequenced by Source Bioscience (Source Bioscience, Nottingham, UK) to confirm correct amplification and the number of polyQ repeats.

PolyQ statistical analysis: The lengths of *AR1* and *AR2* were combined to give an overall *AR* polyQ repeat tract length. As the *AR* locus is positioned on the X chromosome, males only have one copy of the gene while females have two. Therefore, the lengths of *AR1* and *AR2* were simply summed for males, while the mean of the summed lengths for both alleles was used for females. For *NCOA3*, the mean length of the polyQ tract for both alleles was determined for all individuals. Student’s *t*-tests were performed using R statistical software [133] to establish any differences in the length of the polyQ repeat in *AR* and *NCOA3* between affected and unaffected control groups. The individuals included in the unaffected groups were established by the age restrictions as determined previously and as determined in the results. The groups were further split by sex and analysed by ANOVA using R statistical software [133].

## 5. Conclusions

Variations in the *AR* and *NCOA3* genes polyQ repeat tracts are not associated with the development of canine OSA in IWHs and Rottweilers. However, this study has identified several potential therapeutic targets for improving the outcomes of dogs with OSA. Drugs targeting some of the genes identified already exist and thus could be rapidly investigated for an effect in OSA. Additional work is required to determine the functional/mechanistic importance of the difference in expression between tumor and non-tumor tissue and the potential for utilising spontaneous OSA in dogs to improve understanding of the disease in people.

## Figures and Tables

**Figure 1 cancers-12-02405-f001:**
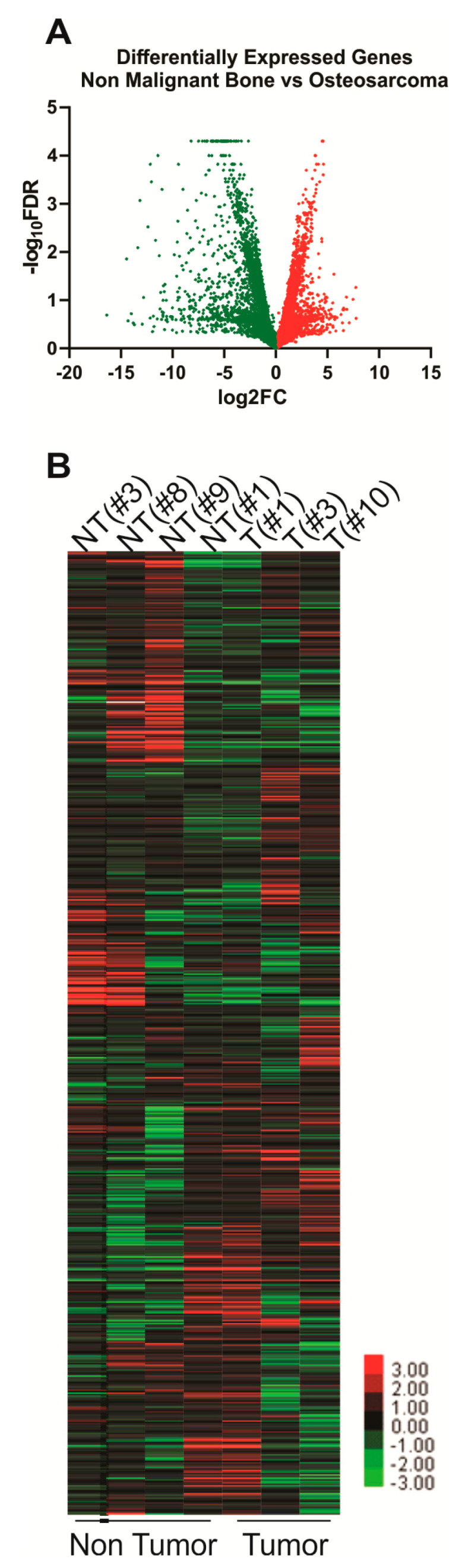
Differentially expressed genes were identified in OSA relative to non-malignant bone. RNA sequencing was used to identify differentially expressed genes expressed lower (green) and higher (red) in OSA tumor specimens relative to non-malignant bone specimens. A volcano plot (**A**) and unsupervised hierarchical clustering was used to display the identified differentially expressed genes (**B**).

**Figure 2 cancers-12-02405-f002:**
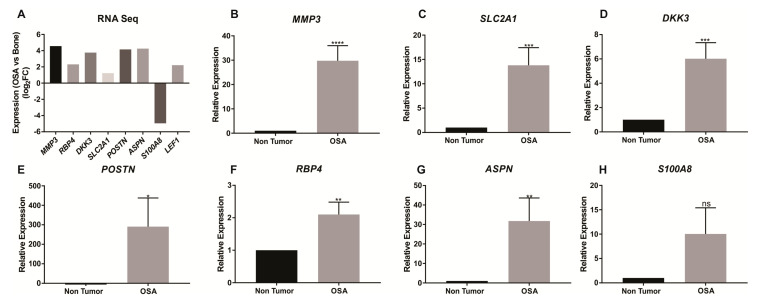
qRT-PCR validation of RNA expression comparing osteosarcoma tumor tissue expression with non-tumor bone tissue expression. Differential expression of key genes was identified by RNAseq (**A**). Results from *MMP3*, *SLC2A1*, *DKK3*, *POSTN*, *RBP4,* and *ASPN* (**B**–**H**) were consistent with the RNAseq results, however, no significant differences were observed in *S100A8* (**H**). Significance levels * *p* < 0.05, ** *p* < 0.01, *** *p* < 0.001, **** *p* < 0.0001, ns = not significant.

**Figure 3 cancers-12-02405-f003:**
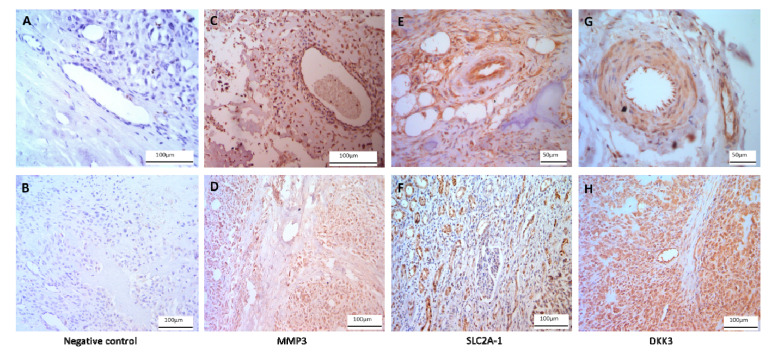
Protein expression of MMP3, SLC2A1, and DKK3 in osteosarcoma tumor. Negative control immunohistochemistry tissues received no primary antibody and were incubated in fetal calf serum instead (**A**,**B**). Immunohistochemistry confirmed protein expression of MMP3 (**C**,**D**), SLC2A1 (**E**,**F**), and DKK3 (**G**,**H**) in osteosarcoma tissue (*n* = 16). Scale bar represents 50 µm (**E**,**G**) and 100 µm (**A**–**D**,**F**,**H**).

**Figure 4 cancers-12-02405-f004:**
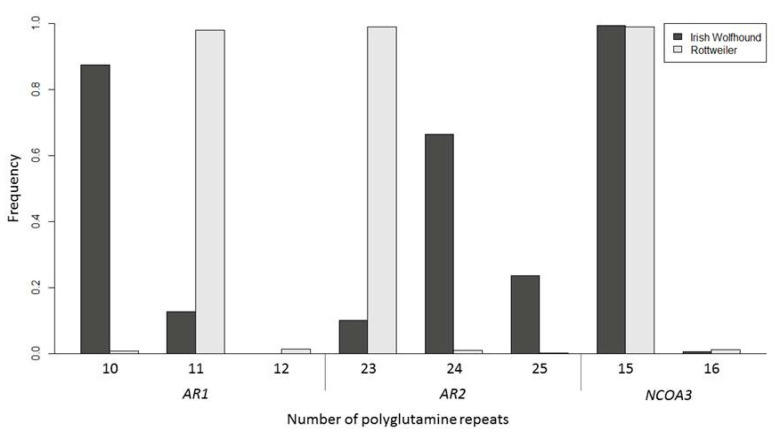
PolyQ repeat allele frequencies for AR1, AR2, and NCOA3. Irish Wolfhounds (*n* = 379) and Rottweilers (*n* = 1019).

**Table 1 cancers-12-02405-t001:** Top 25 genes expressed lower in osteosarcoma relative to non-malignant bone.

Gene_ID	Gene	Log2 Fold Change	*p* Value
ENSCAFG00000007134	*MYBPC1*	−14.4441	0.01395
ENSCAFG00000001705	*MB*	−13.3627	0.0081
ENSCAFG00000013875	*MYL1*	−13.1689	0.00085
ENSCAFG00000008253	*ACTA1*	−12.3844	0.003
ENSCAFG00000011103	*NRAP*	−12.1537	0.00015
ENSCAFG00000005717	*ENSCAFG00000005717*	−12.0616	0.00035
ENSCAFG00000014281	*PYGM*	−11.6411	0.0057
ENSCAFG00000011619	*MYH7*	−11.4064	0.0001
ENSCAFG00000002620	*TNNT1*	−11.3052	0.0119
ENSCAFG00000001016	*TRDN*	−11.034	0.0005
ENSCAFG00000009333	*CESDD1*	−10.7269	0.04655
ENSCAFG00000005896	*RYR1*	−9.83487	0.01135
ENSCAFG00000009404	*TNNC1*	−9.61599	0.0317
ENSCAFG00000015475	*DES*	−9.54008	0.0062
ENSCAFG00000005358	*HHATL*	−9.49779	0.0023
ENSCAFG00000008707	*CKMT2*	−9.43202	0.00015
ENSCAFG00000017254	*ATP2A1*	−9.34349	0.01225
ENSCAFG00000001140	*MYOT*	−9.08389	0.0191
ENSCAFG00000012927	*ALPK3*	−9.01117	0.0005
ENSCAFG00000005343	*KLHL40*	−8.96012	0.0234
ENSCAFG00000002982	*MYBPC2*	−8.66855	0.01675
ENSCAFG00000012432	*MYOZ2*	−8.62006	0.00415
ENSCAFG00000011638	*SPTA1*	−8.56014	0.00135
ENSCAFG00000001161	*OBSCN*	−8.44333	0.01045
ENSCAFG00000011418	*HFE2*	−8.31185	0.0051

RNA sequencing was used to identified differentially expressed genes in OSA (osteosarcoma) specimens relative to non-malignant bone. The entire list of 1281 differentially expressed genes is reported in Appendix A.

**Table 2 cancers-12-02405-t002:** Top 25 genes expressed higher in osteosarcoma relative to non-malignant bone.

Gene_ID	Gene	Log2 Fold Change	*p* Value
ENSCAFG00000010484	*PKP2*	3.54183	0.00095
ENSCAFG00000005510	*EOMES*	3.56165	0.04155
ENSCAFG00000012983	*GPR64*	3.66911	0.0009
ENSCAFG00000013448	*COLGALT2*	3.69448	0.0007
ENSCAFG00000007845	*DKK3*	3.76815	0.0013
ENSCAFG00000009722	*TOX3*	3.77935	0.0002
ENSCAFG00000023615	*MMP-12*	3.78547	0.0001
ENSCAFG00000002589	*COL9A1*	3.84351	0.0001
ENSCAFG00000003503	*ARHGEF5*	3.84945	0.0006
ENSCAFG00000005101	*RPSA*	3.86821	0.00135
ENSCAFG00000019918	*FOXF1*	3.8936	0.0005
ENSCAFG00000017309	*RPL4*	3.92678	0.00015
ENSCAFG00000018123	*ARSI*	3.92752	0.0004
ENSCAFG00000007280	*RNF180*	4.12335	0.00015
ENSCAFG00000010205	*ID4*	4.12517	0.0108
ENSCAFG00000006073	*POSTN*	4.15174	0.03365
ENSCAFG00000009756	*ELOVL2*	4.23619	0.00035
ENSCAFG00000002307	*ASPN*	4.25607	0.02175
ENSCAFG00000009135	*SAA1*	4.42549	0.0053
ENSCAFG00000019036	*LIPG*	4.44654	0.0061
ENSCAFG00000016731	*MSX2*	4.46357	0.0337
ENSCAFG00000015063	*MMP3*	4.56236	0.0337
ENSCAFG00000018597	*LAMA1*	4.58776	0.00025
ENSCAFG00000029131	*HAPLN1*	4.61072	0.00015
ENSCAFG00000031443	*EREG*	5.61347	0.02875

RNA sequencing was used to identified differentially expressed genes in OSA specimens relative to non-malignant bone. The entire list of 1281 differentially expressed genes is reported in Appendix A. Expressions of *MMP3, ASPN, DKK3,* and *POSTN* were validated using qRT-PCR.

**Table 3 cancers-12-02405-t003:** KEGG, Panther and Reactome pathway analysis identified as significantly (q < 0.05) enriched in the dataset. Differentially expressed genes associated with canine OSA were used in a comparative search of the human KEGG, Panther and Reactome pathway databases to identify over-represented pathways.

Gene Set	Database	Description	Size	Expect	Ratio	False Discovery Rate (FDR)
hsa05414	KEGG	Dilated cardiomyopathy	90	5.9888	2.6717	0.04748
hsa05410	KEGG	Hypertrophic cardiomyopathy	83	5.523	2.7159	0.04748
hsa04020	KEGG	Calcium signaling pathway	183	12.177	2.053	0.04748
P02746	Panther	Heme biosynthesis	12	0.96911	7.2231	0.001603
R-HSA-397014	Reactome	Muscle contraction	206	13.605	2.7932	0.0075
R-HSA-189451	Reactome	Heme biosynthesis	11	0.72645	11.012	0.00
R-HSA-390522	Reactome	Striated Muscle Contraction	36	2.3775	5.468	0.000151
R-HSA-189445	Reactome	Metabolism of porphyrins	17	1.1227	8.0164	0.00017
R-HSA-109582	Reactome	Hemostasis	620	40.946	1.6363	0.014273
R-HSA-5578775	Reactome	Ion homeostasis	56	3.6983	3.5151	0.018011
R-HSA-5576891	Reactome	Cardiac conduction	141	9.3118	2.3626	0.037904

**Table 4 cancers-12-02405-t004:** Breed and sex ages at diagnosis.

OSA Cases and Diagnosis	IWH (*n* = 379)	Rottweiler (*n* = 1019)
Number of OSA cases (%)	0.53	3.04
Male age of diagnosis (mean, years)	5.34	7.65
Female age of diagnosis (mean, years)	7.00 **	8.50 *

* *p* > 0.05 ** *p* > 0.01. No significant difference between number of affected males and females in each breed.

**Table 5 cancers-12-02405-t005:** Number of *AR* polyglutamine (polyQ) repeats in IWHs and Rottweilers in OSA/non OSA males and females.

Breed	Group	Diagnosis and Age	AR1 Q-Repeat	AR2 Q-Repeat	AR Q-Repeat Number (Mean ± SEM)
IWH	Male	OSA	10.25 ± 0.25 (4)	24.50 ± 0.29 (4)	34.75 ± 0.25 (4)
		Non OSA over 6.5 years	10.05 ± 0.05 (19)	24.31 ± 0.12 (16)	34.31 ± 0.12 (16)
	Female	OSA	10.13 ± 0.13 (8)	24.57 ± 0.13 (7)	34.57 ± 0.13 (7)
		Non OSA over 9 years	10.04 ± 0.04 (13)	24.19 ± 0.16 (13)	34.23 ± 0.15 (13)
Rottweiler	Male	OSA	11.00 ± 0.00 (12)	23.00 ± 0.00 (12)	34.00 ± 0.00 (12)
		Non OSA over 9 years	11.00 ± 0.00 (49)	23.00 ± 0.00 (50)	34.00 ± 0.00 (49)
	Female	OSA	11.03 ± 0.03 (19)	23.00 ± 0.00 (18)	34.03 ± 0.03 (18)
		Non OSA over 10 years	11.00 ± 0.00 (43)	22.98 ± 0.02 (45)	34.00 ± 0.00 (43)

Numbers represent mean ± SEM (number of samples), OSA = osteosarcoma.

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
