# Peer review of "Molecular Characterisation of Canine Osteosarcoma in High Risk Breeds"

_cancers, 2020, doi:10.3390/cancers12092405_

Round 1

Reviewer 1 Report

In the abstract:

"Pathway analysis of Wiki and KEGG and five ontologies from the biological processes category showed differentially expressed genes." This statement does not make sense what I think the authors are trying to say is Differentially expressed genes were enriched for membership in 5 biological process gene ontologies (GO) as well as WikiPathway and KEGG pathways. 

Table 3 could be a supplement with the text about the factors of interest and their p-values giving enough information. 

The first sentences of section 2.2 could be rewritten. I suggest: Validation of 8 differentially-expressed genes identified by RNAseq was performed by qRT-PCR. Expression for MMP3, MMP2, SLC2A1,DDK3, POSTN, RBP4 and ASPN was confirmed to be higher in the non-tumor tissues. Differential expression of S100A8 was not confirmed by qRT-PCR. 

I also don't like that the qRT-PCR data is split across two figures. I would merge them into a single figure with two subsections. Part A would contain the qRT-PCR data and b would include the IHC. 

I think a table summarizing the findings in section 2.3 would be useful. 

Figure 3 would be better as a table similar to table 4 or as boxplots. 

In the discussion you imply that you performed qRT-PCR validation on the transcription factors. It is unclear from the rest of the paper if you did qRT-PCR validation of the expression levels of the transcription factors of interest. If so I would include those plots with those 8 other validated genes. As the text reads now your list of DE genes id by RNAseq is enriched for genes that are targets of these factors and the 8 genes you did perform qRT-PCR on are targets of one or more of these factors. 

Line 333 "Significantly enhanced gene ontologies" should be significantly enriched gene ontologies. 

Information about how WebGestalt was used for pathway and transcription factor enrichment analysis should be included in the methods. 

Primer information for qRT-PCR and antibody information for IHC should be included. 

 I would include a table of the full differential gene expression analysis results as well as FPKM tables from Cufflinks. This could be a supplement and/or included with a submission of this data to GEO/SRA 

Author Response

1) In the abstract: "Pathway analysis of Wiki and KEGG and five ontologies from the biological processes category showed differentially expressed genes." This statement does not make sense what I think the authors are trying to say is Differentially expressed genes were enriched for membership in 5 biological process gene ontologies (GO) as well as WikiPathway and KEGG pathways.

Thank you we have made this change as suggested.

2) Table 3 could be a supplement with the text about the factors of interest and their p-values giving enough information.

We have now added additional information and presented this in Supplemental Tables. We simplified Table 3 which now shows the enriched KEGG, Panther and Reactome pathway analysis significant results. In the discussion we now provide a brief explanation why assessing distinct pathway databases provides the most comprehensive interpretation of the differentially expressed genes.

3) The first sentences of section 2.2 could be rewritten. I suggest: Validation of 8 differentially-expressed genes identified by RNAseq was performed by qRT-PCR. Expression for MMP3, MMP2, SLC2A1,DDK3, POSTN, RBP4 and ASPN was confirmed to be higher in the non-tumor tissues. Differential expression of S100A8 was not confirmed by qRT-PCR.
Thank you we have made this change as suggested.

4) I also don't like that the qRT-PCR data is split across two figures. I would merge them into a single figure with two subsections. Part A would contain the qRT-PCR data and b would include the IHC.

Thank you. As requested we have now combined the qRT-PCR data and IHC data into two separate figures (Figures 1 and 2).

5) I think a table summarizing the findings in section 2.3 would be useful.

We have inserted a table as suggested (now Table 4). The second part of section 2.3 is also summarised in Table 5 so we have written this in.

6) Figure 3 would be better as a table similar to table 4 or as boxplots.

These are pure frequencies so we cannot do boxplots. As we now have a large number of tables in the manuscript we propose that retaining this data in graphical format will aid the clarity of the manuscript and assist in interpretation of the results.

7) In the discussion you imply that you performed qRT-PCR validation on the transcription factors. It is unclear from the rest of the paper if you did qRT-PCR validation of the expression levels of the transcription factors of interest. If so I would include those plots with those 8 other validated genes. As the text reads now your list of DE genes id by RNAseq is enriched for genes that are targets of these factors and the 8 genes you did perform qRT-PCR on are targets of one or more of these factors.

Our apologies for working this section poorly. We have now rewritten this section in both the results and discussion to make it clear that these are predicted transcription factor networks identified from the differentially expressed gene lists.

8) Line 333 "Significantly enhanced gene ontologies" should be significantly enriched gene ontologies.

Thank you we have made this change as suggested. 

9) Information about how WebGestalt was used for pathway and transcription factor enrichment analysis should be included in the methods.

This has been added in section 4.2 and restated in the results section.

10) Primer information for qRT-PCR and antibody information for IHC should be included.

Full antibody information including product name, number, company and dilution is given in section 4.4 (methods).

For qRT-PCR we used TaqMan assays and we have now added the assay identifiers, company details and a reference in section 4.3 (methods) as requested.

11) I would include a table of the full differential gene expression analysis results as well as FPKM tables from Cufflinks. This could be a supplement and/or included with a submission of this data to GEO/SRA

We have added 3 supplemental tables including the differentially expressed genes. The data submitted the data to the NCBI-GEO database (accession # GSE155646). Reviewer access will be made available upon request and the data will be made publicly available upon publication of this manuscript.  

Reviewer 2 Report

Overall this paper is very interesting and further expands our genetic understanding of canine OSA in high risk breeds. This has ramifications for the successful utilisation of naturally occurring canine OSA as a model for human OSA. 

General comments:

1) The paper is well written and logically presented;

2) Collecting and processing 1019 buccal swabs is an impressive achievement -well done; and

3) I think that all sub-figures should be enumerated and that each sub-figure should be referred to in the corresponding legend.

Specific comments:

1) Consider adding heat maps and venn diagrams to represent the data reported in section 2.1. These diagrams would complement the information in the text.

2) Add a sentence to discuss why gene ontology analysis, KEGG pathways and Wiki pathway where all used and what are the differences and similarities between the programs. Would you expect them to pick out different results? 

3) Consider putting figure 1 and 2 together. The figure legend in figure 1 should be expanded to include details of what the negative control is; and number of dogs used for IHC. 

4) Figure 3 - why are there no error bars included on this figure? 

5) Will the RNAseq data be publicly available? If so, then details of where to find the data should be included in the materials and methods. 

Author Response

Overall this paper is very interesting and further expands our genetic understanding of canine OSA in high risk breeds. This has ramifications for the successful utilisation of naturally occurring canine OSA as a model for human OSA

General comments:
1) The paper is well written and logically presented; Collecting and processing 1019 buccal swabs is an impressive achievement -well done.

Thank you.

2) I think that all sub-figures should be enumerated and that each sub-figure should be referred to in the corresponding legend.

This has been changed as requested.

Specific comments:
3) Consider adding heat maps and venn diagrams to represent the data reported in section 2.1. These diagrams would complement the information in the text.

We have now provided a heatmap representation of the RNAseq data as requested (Figure 1).

4) Add a sentence to discuss why gene ontology analysis, KEGG pathways and Wiki pathway where all used and what are the differences and similarities between the programs. Would you expect them to pick out different results?

We have added discussion around this in the ‘discussion’ section now, thank you.

5) Consider putting figure 1 and 2 together. The figure legend in figure 1 should be expanded to include details of what the negative control is; and number of dogs used for IHC.

qRT-PCR data from the two figures (originally Figures 1 and 2) have been combined and the qRT-PCR and IHC data are now two separate figures (Figures 1 and 2). The negative control and number of dogs used for IHC has been added into the figure legend.

6) Figure 3 - why are there no error bars included on this figure?

These are number of dogs, therefore there are no error bars, as they are exact numbers.

7) Will the RNAseq data be publicly available? If so, then details of where to find the data should be included in the materials and methods.

The data submitted the data to the NCBI-GEO database (accession # GSE155646).